# Pulsed Electric Field-Assisted Enzymatic and Alcoholic–Alkaline Production of Porous Granular Cold-Water-Soluble Starch: A Carrier with Efficient Zeaxanthin-Loading Capacity

**DOI:** 10.3390/foods12173189

**Published:** 2023-08-24

**Authors:** Huanqing Lei, Zhongjuan Liao, Langhong Wang, Xinan Zeng, Zhong Han

**Affiliations:** 1School of Food Science and Engineering, South China University of Technology, Guangzhou 510641, China; leihuanqing@hotmail.com (H.L.); liaozj0315@163.com (Z.L.); 2Guangdong Provincial Key Laboratory of Intelligent Food Manufacturing, Foshan University, Foshan 528225, China; wlhong@fosu.edu.cn (L.W.); xazeng@scut.edu.cn (X.Z.); 3Preparatory Office of Yangjiang Applied Undergraduate College, Yangjiang 529500, China; 4Overseas Expertise Introduction Center for Discipline Innovation of Food Nutrition and Human Health (111 Center), Guangzhou 510641, China

**Keywords:** granular cold-water-soluble starch, pulsed electric field, porous starch, zeaxanthin

## Abstract

In this study, porous starch was modified using pulsed electric field (PEF) pretreatment and alcoholic–alkaline treatment to prepare porous granular cold-water-soluble starch (P-GCWSS). The soluble porous starch has high adsorption capability and high cold water solubility, allowing effective encapsulation of zeaxanthin and improving zeaxanthin’s water solubility, stability, and bioavailability. The physical and chemical properties of GCWSS and complex were investigated using scanning electron microscopy, Fourier transform infrared spectroscopy, and X-ray diffraction. The results showed that the cold water solubility of the pulsed electric field-treated porous granular cold-water-soluble starch (PEF-P-GCWSS) increased by 12.81% compared to granular cold-water-soluble starch (GCWSS). The pulsed electric field treatment also increased the oil absorption of PEF-P-GCWSS was improved by 15.32% compared to porous granular cold-water-soluble starch (P-GCWSS). PEF-P-GCWSS was effective in encapsulating zeaxanthin, which provided a good protection for zeaxanthin. The zeaxanthin-saturated solubility in water of PPG–Z was increased by 56.72% compared with free zeaxanthin. The zeaxanthin embedded in PEF-P-GCWSS was able to be released slowly during gastric digestion and released rapidly during intestinal digestion.

## 1. Introduction

Modified starch with pore structures on the surface or throughout the entire starch particulate is called porous starch. The physicochemical properties of porous starch are closely related to its pore structures. Porous starch has excellent biodegradability, biocompatibility, and safety [1]. In addition, high porosity, large specific surface area, low particle density, and strong adsorption performance are also important advantages of porous starch [2]. The special pore structure of porous starch is associated with the increased specific surface area and pore volume, resulting in significant changes in its adsorption performance [3]. Porous starch can not only firmly adsorb substances onto the inner wall of the starch particles through the capillary action of internal pores; it can also serve as an encapsulation wall material to counteract the unstable properties of substances such as sensitivity to light, temperature, and oxygen [4]. Due to its abundant availability, low cost, and wide applicability, it is widely used in the food, medicine, agricultural products, cosmetics, paper industries [5]. Piloni et.al. [6] prepared porous starch using α-amylase and glucoamylase hydrolysis, creating porous starch adsorbent for chia oil. Porous starch offered good oxidative protection to chia oil. The work of Wang et.al. [7] showed that porous gelatinized corn starch may be used as an effective adsorbent in order to improve the utilization efficiency of grape seed proanthocyanidins and maintain their antioxidant activity. Yaiza et.al. [8] developed a system to thermally stabilize probiotic bacteria based on porous starches and used biopolymers as coating materials.

The mechanism of pulsed electric field (PEF) on starch [9] is usually believed to be the charge polarization effect formed by its strong pulse action, which changes the structural properties of starch particles. Under the action of PEF, the charged particles are accumulated on the surface of starch granules, and they generate macroscopic space charges by moving directionally in the starch solution [10]. When the electric field strength reaches a certain value, the outer layer of the starch granules will rupture and disintegrate under the effect of the instantaneous high-voltage discharge. The results of Han et al. [11] showed that PEF treatment can transform the surface of starch granules from smooth to rough and form some pits, which will reduce the crystallinity and enthalpy of pasting of starch granules so that the thermal stability of starch paste will be significantly changed. The high intensity treatment of PEF can disrupt the crystalline and granular structure of starch, block the molecular chain of starch, and reduce the molecular mass of starch. PEF treatment can also increase the rate of chemical modification of starch by boosting the directional movement of charged chemical reagents [12]. In summary, PEF treatment is an effective technical means by which to change the apparent morphology and intrinsic structure of starch, which has a significant application potential in starch modification.

As starch is naturally not easily soluble in water, modification methods are needed to increase its cold water solubility [13]. A modified starch that can be dissolved in cold water to form a paste via modification of its original structure using physical, chemical, or biological enzymatic hydrolysis techniques is called granular cold-water-soluble starch (GCWSS) [14]. Among the physical methods typically employed, the alcoholic–alkaline method is a widely used method for the preparation of GCWSS due to its simplicity in terms of equipment and operation, its high level of safety, and the controllable quality of the product [15]. GCWSS can be quickly dissolved in cold water to form starch paste with good properties, such as high homogeneity and stability, with a transparent appearance and suitable viscosity. It has superior water retention, thickening, and emulsifying effects, as well as good high temperature tolerance and freeze–thaw stability [16]. In recent years, GCWSS has been applied in the food, agriculture, and textiles industries [17]. GCWSS has a certain degree of adsorption due to the single-helix structure [18] formed during the processing of GCWSS, allowing it to easily combine with organic substances to form V-complexes. GCWSS starches can also be used to encapsulate materials such as the encapsulation of ethylene gas [19] and cinnamaldehyde [20]. The results of starch modification are usually characterized using techniques such as scanning electron microscopy (SEM), Fourier transform infrared spectroscopy (FTIR), and X-ray diffraction (XRD), which were used to characterize the morphology, short-range crystalline structure, and long-range crystalline structure of starch.

Zeaxanthin (C_40_H_56_O_2_) is a fat-soluble yellow powder or oil. Zeaxanthin has good antioxidant activity, which is important for human physiological health. However, zeaxanthin can only be obtained by the body through the diet as it cannot be synthesized by the body. Zeaxanthin protects visual function [21], reduces atherosclerosis to alleviate cardiovascular disease, prevents DNA damage to epithelial cells via ultraviolet radiation, and maintains human cognitive function [22,23]. Zeaxanthin is limited in application due to its poor water solubility and susceptibility to environmental influences [24] (e.g., heat, light, oxygen, metals, enzymes, etc.). The antioxidant property, low bioavailability, and color of zeaxanthin will be affected [25]; therefore, it is necessary to find a low-cost and effective embedding matrix in order to realize the protection and solubilization of zeaxanthin, which can extend the storage period of zeaxanthin and improve its range of application.

In summary, porous starch has an excellent adsorption capacity due to its many pores and large specific surface area. However, its low modification efficiency and poor water solubility limit the application and development of porous starch. Pulsed electric field is a physical technique that can effectively modify the properties of porous starch. In order to solve the problem of poor water solubility, porous starch was modified via the alcohol–alkali method to obtain porous granular cold-water-soluble starch. Therefore, the aim of this study was to develop a granular cold-water-soluble porous starch with high adsorption properties, which can broaden the application of porous starch, and to improve the bioavailability of active substances. In order to solve the disadvantages of low efficiency and high energy loss in the enzymatic preparation of porous starch, PEF pretreatment was used to improve the enzymatic efficiency. To water soluble porous starch, the alcoholic-alkaline method was used to modify the porous starch and to improve the cold water solubility characteristics. The water-soluble porous starch was then evaluated as a matrix, efficiently encapsulating zeaxanthin to improve its solubility in water and translating it to improve its stability and slow-release ability.

## 2. Materials and Methods

### 2.1. Material and Chemicals

Food-grade corn starch (CS) was purchased from Shanghai Macklin Biochemical Co., Ltd. (Shanghai, China). Analytical-grade potassium chloride, glacial acetic acid, sodium hydroxide, sodium chloride, potassium dihydrogen phosphate, DMSO, and anhydrous sodium acetate were purchased from Shanghai Macklin Biochemical Co., Ltd. Glucose amylase was purchased from Sigma Aldrich CO. (St. Louis, MO, USA). α-amylase, pancreatin, and pepsin were purchased from Shanghai Macklin Biochemical Co., Ltd. Analytical-grade anhydrous ethanol purchased from Tianjin Fuyu Fine Chemical Co., Ltd. (Tianjin, China). Hydrochloric acid was purchased from Guangdong Guangshi Reagent Technology Co., Ltd. Soybean oil was purchased from COFCO Fulinmen Food Marketing Co., Ltd. (Nantong, China).

### 2.2. Preparation of Different Starch Samples

A simple procedure for sample preparation is shown in Figure 1 with the following steps:(1)Pulsed electric field-treated corn starch (PEF-CS): Corn starch (CS) was dissolved in deionized water with a concentration of 15% (*m*/*m*). Potassium chloride solution (0.5 mol/L) was added to the starch dispersion to increase the conductivity of the starch dispersion to 150 ± 5 μS/cm. The starch dispersion was pumped into the PEF treatment equipment [11] at a constant flow rate of 2 mL/s. The corn starch was pretreated and modified under the electric field intensity of 12 kV/cm and an effective treatment time of 18 ms. The starch emulsion was centrifuged to obtain a precipitated sample, washed 3 times with deionized water, vacuum filtered, dried, and ground into powder to obtain pulsed electric fields-pretreated corn starch (PEF-CS).(2)Porous starch (PS): Corn starch was dissolved in acetic acid sodium acetate (Hac-NaAc, pH = 5.0) buffer solution with a concentration of 30% (*m*/*m*). Starch dispersion was stirred and activated for 30 min under water bath conditions at 50 °C. The enzymatic hydrolysis method was slightly modified based on the method proposed by Chen et al. [26]. α-amylase and glucose amylase were prepared as composite enzymes in a ratio of 1:1. The sample was enzymatically hydrolyzed for 4.5 h with an enzyme concentration of 2.0%. The precipitates were obtained by centrifuging the sample at 8000 rpm/min for 20 min and were washed 4 times with deionized water and anhydrous ethanol, respectively. After vacuum filtration, drying, and grinding of the precipitated sample, porous starch samples (PS) were obtained.(3)Pulsed electric field-treated porous starch (PEF-PS): Similar to the preparation method of PS, porous starch assisted by pulsed electric fields (PEF-PS) was obtained by replacing corn starch with PEF-CS.(4)Granular cold-water-soluble starch (GCWSS): Corn starch (CS) was modified with the alcoholic–alkaline method [27,28]. Corn starch was mixed with 80% ethanol to form a starch dispersion and stirred evenly in a 25 °C water bath. An amount of 4.8 wt% sodium hydroxide was added to the system, and the pH was adjusted to neutral after a reaction time of 15 min. The sample was washed four times with anhydrous ethanol, vacuum filtered, and dried to a constant weight to obtain granular cold-water-soluble starch (GCWSS).(5)Pulsed electric field-treated granular cold-water-soluble starch (PEF-GCWSS): Similar to the preparation method of GCWSS, pulsed electric fields-pretreated granule cold-water-soluble starch (PEF-GCWSS) was obtained by replacing corn starch with PEF-CS.(6)Porous granular cold-water-soluble starch (P-GCWSS): Similar to the preparation method of GCWSS, granule cold-water-soluble porous starch (P-GCWSS) [15] was obtained by replacing corn starch with PS.(7)Pulsed electric field-treated porous granular cold-water-soluble starch (PEF-P-GCWSS): Similar to the preparation method of GCWSS, granular cold-water-soluble porous starch assisted by pulsed electric fields (PEF-P-GCWSS) was obtained by replacing corn starch with PEF-PS.
Figure 1Preparation of different starch samples.
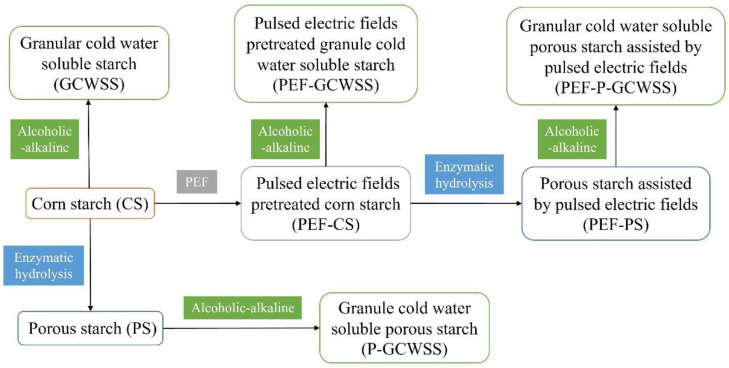



### 2.3. Preparation of Starch–Zeaxanthin Composites

The different samples (CS, PEF-PS, P-GCWSS, and PEF-P-GCWSS) were dispersed in anhydrous ethanol with a concentration of 10% (m/m). Zeaxanthin solution (20 mg zeaxanthin in 10 mL anhydrous ethanol) was added to the starch dispersion and incubated for 2 h under magnetic stirring at 600 r/min [29,30]. The unencapsulated zeaxanthin was washed with anhydrous ethanol. The precipitate was oven-dried at 40 °C for 2 h to obtain the original starch–zeaxanthin composites (CS–Z), PEF-assisted preparation of porous starch–zeaxanthin composites (PPS–Z), granular cold-water-soluble porous starch–zeaxanthin composites (PG–Z), and PEF-assisted preparation of granular cold-water-soluble porous starch–zeaxanthin composites (PPG–Z).

### 2.4. Cold Water Solubility and Oil-Adsorption Capacity

Cold water solubility was determined according to Chen [15] with some modification. Starch dispersion with a concentration of 1% (*m*/*v*) was stirred in a water bath at 25 °C for 20 min. After centrifugation for 20 min (3000 rpm/min), the supernatant was transferred into weighed flat bottles and dried in an oven at 105 °C for 6 h to constant weight. 

The equation for cold water solubility (*CWS*) is as follows:(1)$$CWS\left(\%\right)=\frac{m_{1}}{m_{0}}\times 100,$$
where *m*_0_ (g) is the mass of starch sample; and *m*_1_ is the mass of dried sample.

Oil-adsorption capacity was measured according to Fang [31] with some modification. The mass of the centrifuge tube containing the starch sample (1.0 g) is m_1_. A total of 10 mL of soybean oil was added to the centrifuge tube and mixed at 25 °C for 30 min; the floating soybean oil was removed by centrifugation at 8000 rpm/min for 20 min. The total mass of the remaining precipitated sample and the centrifuge tube is recorded as m_2_. The equation for oil-adsorption capacity (*OA*) is as follows:(2)$$OA\left(\%\right)=\frac{m_{3}-m_{2}}{1.0}\times 100,$$
where *m*_2_ (g) is the mass of the centrifuge tube containing the starch sample (1.0 g); and *m*_3_ is the total mass of the precipitated sample and the centrifuge tube.

### 2.5. Scanning Electron Microscopy (SEM) 

The apparent morphology of the samples was observed by scanning electron microscope [32] (EVO18, Zeiss Co., Ltd., Oberkochen, Germany). A small amount of the sample was uniformly adhered to the conductive adhesive, and vacuum gold coating was carried out for 30 s. The surface optical properties of starch particles were observed at a magnification of 3000 (starch samples) or 5000 (starch–Zeaxanthin composites) at the acceleration voltage of 5 kV.

### 2.6. Fourier Transform Infrared Spectroscopy (FTIR)

Fourier transform infrared spectroscopy (Spectrum 3, PerkinElmer Co., Ltd., Waltham, MA, USA) was used to characterize the short-range crystal structure of starch samples [33]. All samples were scanned between the 4000–650 cm^−1^ wavenumber range at a resolution of 4 cm^−1^.

### 2.7. X-ray Diffraction (XRD)

X-ray diffraction (D8 ADVANCE, Bruker Co., Ltd., Karlsruhe, Germany) was used to characterize the long-range crystalline structure of the sample at 40 kV and 40 mA [32]. The diffraction angle ranged from 4° to 40° (2θ) with a scan rate of 0.02°/s. The relative crystallinity (R) was calculated using MD Jade 5.0 software, as follows:(3)$$R\left(\%\right)=\frac{crystalline\;area}{total\;diffractogram\;area}\times 100.$$

### 2.8. Zeaxanthin Adsorption Capacity and Encapsulation Efficiency

The zeaxanthin content and encapsulation efficiency were determined according to the previous methods [30] with minor modifications. The composites were completely dispersed in dimethyl sulphoxide (DMSO) by vortexing. After centrifugation at 3000 r/min for 10 min, the supernatant was diluted ten times and its absorbance measured at the 446 nm wavelength. The zeaxanthin adsorption capacity (*AC*, mg/g) was calculated with the following equation:(4)$$AC\left(mg/g\right)=\frac{X\left(\mu g/mL\right)\times V\left(mL\right)\times 10}{0.01\times 1000},$$
where *X* (μg/mL) is the zeaxanthin concentration calculated corresponding to the absorbance-concentration zeaxanthin standard curve; and *V* (mL) is the volume of supernatant.

The encapsulation efficiency (*ER*, %) was calculated with the following equation:(5)$$ER=\frac{AC\left(mg/g\right)\times SM\left(g\right)}{ZM\left(mg\right)},$$
where *SM* (g) is the weight of the given composite obtained; *AC* (mg/g) is the zeaxanthin content of the sample; and *ZM* (mg) is the total weight of zeaxanthin added.

### 2.9. Determination of Saturated Solubility

Excessive samples were added to deionized water (5 mL) and incubated at 37 °C for 48 h. The supernatant was collected and diluted after centrifugation at 5000 r/min for 20 min. The absorbance of the solution was measured using a UV spectrophotometer at the 446 nm wavelength. The saturated solubility [34] (*SS*, μg/mL) was calculated with the following equation:(6)$$SS=\frac{SM}{V}\times 100,$$
where *SM* (μg) is the content of zeaxanthin in the saturated solution; and *V* (mL) is the volume of the saturated solution.

### 2.10. Storage Stability

The storage stability (light and temperature) of the sample was determined according to the method reported in [35] with minor modifications. The samples were kept in the light and dark environment at room temperature for 7 days, respectively. Alternatively, the samples were kept dark for 7 days at 4 °C and 20 ° C, respectively. For the analysis, the samples (10.00 mg) were dispersed in DMSO (5 mL). The absorbance of zeaxanthin in the sample was determined at 446 nm via UV spectrophotometer to assess the impact of light and temperature on the stability of the samples. The retention rate (*RR*, %) of the zeaxanthin was calculated as follows:(7)$$RR\left(\%\right)=\frac{A_{t}}{A_{0}}\times 100,$$
where *A_t_* (mg) is the content of zeaxanthin after *t* day; and *A*_0_ (mg) is the initial zeaxanthin content of the samples.

### 2.11. In Vitro Gastric and Intestine Digestion

Simulated gastric fluid (SGF) and simulated intestinal fluid (SIF) were prepared following the methods of McClements [36] and Marefati [37] with minor modifications. The starch sample (5.00 mg) was dispersed in SGF and SIF and incubated at 37 °C under constant temperature shaking. The digestive juice that collected every 30 min was dispersed in DMSO via vortexing. The supernatant was obtained and diluted after centrifugation at 3000 r/min for 10 min, and the absorbance value of the solution was measured at 446 nm using a UV spectrophotometer. The percentage of the zeaxanthin released (*ZR*, %) was calculated as follows:(8)$$ZR\left(\%\right)=\frac{C_{t}}{C_{0}}\times 100,$$
where *C_t_* (mg) is the content of zeaxanthin released at *t* min, and *C*_0_ (mg) is the zeaxanthin content of the sample.

### 2.12. Statistical Analysis

All the experiments were performed at least in triplicate. Data were analyzed by Duncan’s multiple range test (*p* < 0.05) using the SPSS 17.0 statistical software program (SPSS Incorporated, Chicago, IL, USA).

## 3. Results and Discussion

### 3.1. Physicochemical Characterization of the Starch Samples

#### 3.1.1. Cold Water Solubility and Oil Adsorption Rate Analysis

CS, PEF-CS, PS, and PEF-PS were modified using the alcoholic-alkaline method to prepare granular cold-water-soluble starch (GCWSS), PEF-assisted preparation of granular cold-water-soluble starch (PEF-GCWSS), porous granular cold-water-soluble starch (P-GCWSS), and PEF-assisted preparation of granular cold-water-soluble porous starch (PEF-P-GCWSS), respectively. Figure 2A shows the cold water solubility and oil absorption for the different samples. The cold water solubility of GCWSS is 73.60%, while the cold water solubility of PEF-GCWSS has slightly increased to 78.25%. This slight increase may be due to the looser particle structure which could have bound more easily with water molecules through hydrogen bonding [9,31], which improves the efficiency of the reaction. Compared with GCWSS, P-GCWSS has higher cold water solubility, which may be due to the porous structure of the porous starch. The same results were found in the research of Chen et al. [15], who found that granular cold-water-soluble starch is more easily prepared with porous starch than raw starch; it makes the short-chain amylopectin and amylose in the starch granules easier to dissolve. In addition, the reason for the increase in cold water solubility also includes the increase in the specific surface area of the porous starch hydrolyzed by enzymatic hydrolysis [5], which increases the contact area between starch and reactants. The results show that the cold water solubility of PEF-P-GCWSS is the highest because PEF-PS has more pores and larger specific surface area when compared to CS, PEF-CS, and PS. This shows that PEF-assisted enzymatic hydrolysis is a feasible process for preparing granular cold-water-soluble porous starch.

The oil absorption rates of different samples are shown in Figure 2A. The high oil absorption of GCWSS is related to its loose V-shaped single-helix cavity structure, which gives it a certain ability to adsorb substances [38]. P-GCWSS has a higher oil absorption compared to GCWSS because the unique pore structure of porous starch particles endows it with excellent adsorption performance [39]. PEF has a significant impact on the oil absorption rate of starch. The oil absorption rate of PEF-GCWSS was 10.40% higher than GCWSS, and the oil absorption rate of PEF-P-GCWSS was 15.32% higher than P-GCWSS. The oil absorption rate of PEF-P-GCWSS was significantly higher than other groups because the enzyme is more likely to enter the interior of the starch granule under the action of PEF, contributing to the formation of a pore structure. In summary, the modification technique of PEF pretreatment has a positive effect on enhancing the adsorption properties of starch.

#### 3.1.2. SEM Analysis

Scanning electron microscopy of the different samples is shown in Figure 3. After alcoholic-alkaline modification, the GCWSS showed different degrees of swelling and enlargement but still remained granular. The starch granules may appear viscous because the action of alkali causes the starch granules to swell and become viscous. PEF-GCWSS has deeper particle depressions when compared to GCWSS. Similarly, the changes in the particle structure of P-GCWSS and PEF-P-GCWSS manifested as a shift from a surface-porous morphology to a distorted concave morphology. Nevertheless, pores were still observed on the surface, and some of the particles even had deep cracks on the surface, which was conducive to the adsorption of substances [2]. Compared with P-GCWSS, PEF-P-GCWSS has deeper particle depression and a larger number of pores, which led to stronger adsorption behavior. During the alcoholic-alkaline modification process, the internal organization of starch was ruptured by alkali action, and the starch granules dissolved, further undergoing swelling and twisting deformation. At the same time, the excessive swelling of starch granules was prevented by the ethanol solution, thus maintaining their relatively intact granular morphology. Finally, ethanol was released from the interior of the granule during drying, leading to starch granules with different degrees of wrinkles and surface depressions [40].

#### 3.1.3. Fourier Transform Infrared Spectroscopy (FTIR) Analysis

The Fourier transform infrared spectra of the samples are shown in Figure 2B. The positions of the absorption peaks of GCWSS, PEF-GCWSS, P-GCWSS, PEF-P-GCWSS, and CS consistently indicated that the alcoholic-alkaline treatment is a physical modification [41] that does not produce new functional groups or change the original molecular composition of the starch particles. Compared to CS, the intensity of some characteristic absorption peaks in the infrared spectra of other samples, such as 2930 cm^−1^, 1149 cm^–1^ and 996 cm^−1^ peaks, were weakened to various degrees, indicating that the folding structure inside the starch molecules was destroyed.

#### 3.1.4. X-ray Diffraction (XRD) Analysis

The X-ray diffraction analyses of different samples and corn starch are shown in Figure 2C,D respectively. Corn starch samples exhibit a typical A-type crystalline structure, with diffraction peaks located at 15°, 17°, 18°, and 23°, and a slightly weaker diffraction peak at 20° [11]. GCWSS, PEF-GCWSS, P-GCWSS, and PEF-P-GCWSS all showed characteristic absorption peaks at 20 °, which are typical of type V starch. However, the diffraction peaks disappeared at 15°, 17°, 18°, and 23° as the internal crystal structure of starch particles was modified and was destroyed by the alcoholic-alkaline method, resulting in hydrogen bond breakage between starch particles and improved water solubility. When the starch double-helix structure is dissociated by heating; branched and straight-chain starch combine with alcohol to form V-complexes. When ethanol is removed from the V-complex, the starch is in a sub-stable state and can be dissolved in cold water [15]. This is consistent with the physicochemical properties of the V-shaped starch structure (single-helix structure) dissolved in cold water in previous studies [15,28].

### 3.2. Physicochemical Characterization of the Starch–Zeaxanthin Samples

#### 3.2.1. Analysis of the Zeaxanthin Adsorption Capacity and Encapsulation Efficiency

Both the hollow single helix of GCWSS and the pore structure of porous starch can be used to accommodate small guest molecules. In general, hydrophobic interactions and concentration gradients (diffusion) are the main driving forces [42]. The zeaxanthin adsorption capacity and encapsulation efficiency of the various samples are shown in Figure 4A. CS showed the lowest adsorption capacity and lowest encapsulation efficiency of zeaxanthin, which were 12.94 mg/g and 46.21%, respectively, while the adsorption capacity of PEF-PS for zeaxanthin was 17.74 mg/g, and the embedding rate was 63.36%, which was significantly higher than that of CS. The adsorption capacity of zeaxanthin by P-GCWSS was 18.83 mg/g, and the encapsulation efficiency was 67.25%, while the adsorption capacity and the encapsulation efficiency of zeaxanthin by PEF-P-GCWSS were the highest, which increased by 16.67% when compared with that of P-GCWSS, with amounts of 21.97 mg/g and 78.46%, respectively. The results indicate that PEF-P-GCWSS is an encapsulation material with excellent adsorption performance.

#### 3.2.2. SEM Analysis

Scanning electron microscopy of the various encapsulate zeaxanthin samples is shown in Figure 5. Zeaxanthin exhibited irregular stripes or a flaky crystal-like morphology. The weak adsorption of CS to zeaxanthin, which is only adsorbed on the surface, is due to the fact that it has only a small number of grooves and tiny pores. The adsorption of zeaxanthin by PEF-PS particles is stronger due to the presence of deeper grooves and larger pores on their surface. The surface of PPS–Z particles has zeaxanthin fragments filling some of the pores. The P-GCWSS particles showed a twisted and concave porous structure, and some of the particles had deep cracks on the surface, so their adsorption of zeaxanthin was strong. Zeaxanthin fragments could be observed on the surface and the cracks of some of the PG–Z particles, indicating that zeaxanthin could be adsorbed inside the P-GCWSS particles. The PEF-P-GCWSS particles also showed a twisted and concave porous structure, with a deeper degree of concavity and a larger number of pores; these are features conducive to the adsorption of a large amount of zeaxanthin. Most of the pores on the surface of the PPG–Z particles could be clearly seen to be filled with zeaxanthin fragments, which indicated that zeaxanthin had been adsorbed in the interior of the PEF-P-GCWSS particles.

In summary, zeaxanthin was successfully encapsulated and adsorbed into the interior of the starch granules through the cracks, pores, and other channels. The results of the scanning electron micrographs were consistent with those of Figure 4A, indicating that PEF-P-GCWSS had a stronger adsorption capacity and the best encapsulating effect on zeaxanthin compared with PEF-PS and P-GCWSS.

#### 3.2.3. Fourier Transform Infrared Spectroscopy (FTIR) Analysis

The Fourier transform infrared (FTIR) spectra of various encapsulated samples, original starch, and zeaxanthin are shown in Figure 4B. The absorption peaks of zeaxanthin around 3234 cm^−1^ represent the O-H telescopic vibration. Values of 3039 cm^−1^ and 1569 cm^−1^ represent the C–H telescopic vibration in C=C. Around 2923 cm^−1^ is the C-H telescopic vibration in -CH_3_. The value of 2853 cm^−1^ is the C–H telescopic vibration in -CH_2_. Around 1441 cm^−1^ and 1361 cm^−1^ are the C–H in-plane bending vibration peaks in -CH_2_, and 962 cm^−1^ is the C–H out-of-plane bending vibration in C=C [43,44]. The positions of the absorption peaks of PEF-PS, P-GCWSS, PEF-P-GCWSS, and those of CS remained basically the same, while the infrared spectra of the different inclusion samples showed both the absorption peaks of CS and part of the absorption peaks of zeaxanthin with some shifts. A small absorption peak near 965 cm^−1^ indicated the presence of zeaxanthin [45] in the composites and the intermolecular interaction between starch and zeaxanthin.

#### 3.2.4. X-ray Diffraction (XRD) Analysis

The X-ray diffractions of the various encapsulated zeaxanthin samples are shown in Figure 4C. Zeaxanthin showed sharp and strong diffraction peaks at 2θ angles of 7.0°, 13.3°, 17.5°, and 20.6°, respectively, with the strongest peak at 2θ of 20.6°, which indicated that the natural zeaxanthin is a typical crystalline morphology, which leads to the inability of zeaxanthin to dissolve in water [43]. The characteristic diffraction peaks of the encapsulation samples disappeared, and CS–Z, PPS–Z, and CS, PEF-PS showed typical A-type crystalline structures, as well as PG–Z, PPG–Z, and P-GCWSS. The PEF-P-GCWSS showed V-type crystalline structures, which indicated that the zeaxanthin was successfully adsorbed and encapsulated. The existence of zeaxanthin in an amorphous state within the pores can enhance the water solubility of zeaxanthin, which, in turn, has excellent dispersion and bioavailability.

### 3.3. The Saturated Solubility Analysis

The saturated solubility of the various encapsulated zeaxanthin samples was shown in Figure 4D. The saturated solubility of zeaxanthin in water was 9.22 μg/mL, which was extremely insoluble. However, the saturated solubility of zeaxanthin was increased to different degrees after starch encapsulation. In particular, the saturation solubility of the PEF-P-GCWSS-embedded zeaxanthin composites was increased by 56.72% to 14.45 μg/mL, which indicated that the encapsulation by PEF-P-GCWSS was able to improve the aqueous solubility of zeaxanthin to a certain extent.

The water-soluble states of the various encapsulated zeaxanthin samples are shown in the inset of Figure 4D. Since zeaxanthin is extremely difficult to dissolve in water, more zeaxanthin can be clearly seen adhering to the wall of the container in inset A. A small amount of insoluble material can be seen adhering to the wall of the container after dissolving in water for both CS–Z and PPS–Z, and the color distribution of the solution is not uniform, which is mostly shown at the bottom and at the top. While GCWSS was soluble in cold water at room temperature, PG–Z and PPG–Z did not have obvious adherents after dissolving in water, and the color distribution was more homogeneous, which indicated better dissolution in water. In conclusion, PPG–Z can be used in convenience foods by utilizing its water-soluble property.

### 3.4. Storage Stability Analysis

Zeaxanthin has limited industry application because it is susceptible to losses caused by external environmental influences during transportation, storage, and processing [46]. The effects of light on the storage stability of the various encapsulated zeaxanthin samples are shown in Figure 6A,B. The retention rates of zeaxanthin were higher under no-light storage conditions when compared to storage conditions with light, which indicated that no-light is beneficial for the storage of zeaxanthin. The effect of temperature on the storage stability of the various encapsulated zeaxanthin samples is shown in Figure 6C,D. The retention rate of zeaxanthin at 4 °C was higher than that at 20 °C, which indicated that low temperature is good for the storage of zeaxanthin. The retention rates of zeaxanthin of the composites during storage were all significantly higher than those of free zeaxanthin, indicating that the starch-based embedding of zeaxanthin can enhance the resistance of zeaxanthin to light and temperature, thus improving the storage stability of zeaxanthin. The retention of zeaxanthin during storage was significantly higher than that of free zeaxanthin in all of the samples, which indicated that starch, as an encapsulation material, can improve the storage stability of zeaxanthin because of its ability to enhance the resistance to light and temperature. The retention rates of zeaxanthin under the same storage conditions were different for different composites. Specifically, there was an increasing retention rate observed between the samples (CS–Z < PPS–Z < PG–Z < PPG–Z), indicating that the encapsulation of zeaxanthin by porous starch and granular cold-water-soluble porous starch was mainly aggregated in the inner part of the granules, which led to the enhancement of zeaxanthin retention due to the strong adsorption capacity of its pores, thus prolonging the shelf-life of zeaxanthin. The highest retention of zeaxanthin in PPG–Z was 93.09% within 7 days at 4 °C and protected from light. This is an improvement of 15.65% compared to the free zeaxanthin under the same conditions. This observation is related to its highest adsorption performance, where more and deeper pores can form a better encapsulation for zeaxanthin. In summary, PEF-P-GCWSS encapsulation of zeaxanthin is a protection method that can effectively improve the stability of zeaxanthin.

### 3.5. In Vitro Gastric and Intestine Digestion

Zeaxanthin release from different composites under simulated stomach/intestinal conditions is shown in Figure 6E,F. The zeaxanthin in the complex showed a large release in both simulated gastric and intestinal fluids initially. This is because the encapsulation of zeaxanthin by starch is mainly via adsorption. At the initial stage, the zeaxanthin adsorbed on the surface is released relatively quickly. With the extension of time, the zeaxanthin adsorbed in the interior is slowly released. Zeaxanthin release from the different composites was lower in the simulated stomach than in the simulated intestine, suggesting that the degradation of the complexes and the release of zeaxanthin was mainly in the intestine. The increased release is due to the digestion of starch by trypsin in the artificial intestinal fluid, which prompted a rapid release of zeaxanthin, leading to a fuller contact of zeaxanthin with the medium. The stability of zeaxanthin is affected by pH, and zeaxanthin can exist and dissolve more stably at pH values between 4 and 12 [25]. The pH value of the simulated gastric fluid is 2.0, and the pH value of simulated intestinal fluid is 6.5. Zeaxanthin may suffer a certain degree of degradation and destruction under strong acidic conditions.

Free zeaxanthin had the highest cumulative zeaxanthin release rate under simulated gastric/intestinal conditions, and the release rate was significantly higher than that of the complex under simulated gastric conditions, which indicates that starch is an effective carrier for slowing down the release of zeaxanthin in the stomach. The higher rate of zeaxanthin release from CS–Z under simulated gastric/intestinal conditions could be attributed to the weaker adsorption of zeaxanthin by CS. PPS–Z, PG–Z and PPG–Z all showed lower zeaxanthin release rates under simulated stomach/intestinal conditions. The release of zeaxanthin was related to the amount of starch adsorbed and the water solubility of the complexes. The faster release of zeaxanthin from P-GCWSS and PEF-P-GCWSS could be attributed to its ability to dissolve rapidly in cold water. The slower release of zeaxanthin from PEF-PS may be due to its lower solubility in cold water. The rate of zeaxanthin release from PG–Z and PPG–Z was significantly higher in the mock intestine than in the stomach, which may be attributed to the rapid solubility and high digestive sensitivity to enzymes of P-GCWSS and PEF-P-GCWSS. This makes them susceptible to hydrolysis, causing faster release of zeaxanthin. In summary, free zeaxanthin cannot be released slowly under gastric/intestinal conditions, and PEF-P-GCWSS-encapsulated zeaxanthin is an effective strategy for slow release of zeaxanthin in gastric digestion followed by a large amount of zeaxanthin release in intestinal digestion, which improves zeaxanthin absorption and utilization by the human body.

## 4. Conclusions

In the present study, it was shown that PEF-assisted preparation of granular cold-water-soluble porous starch (PEF-P-GCWSS) is an effective encapsulation matrix for zeaxanthin, which improves the solubility, stability, and slow release of zeaxanthin. The PEF-assisted enzymatic method and the alcoholic-alkaline method were used to enhance the adsorption properties and water solubility of porous starch, respectively. The cold water solubility and oil absorption of PEF-P-GCWSS were enhanced. PEF-P-GCWSS was the most effective in encapsulating zeaxanthin, which provided a good protection for zeaxanthin. Zeaxanthin encapsulated in PEF-P-GCWSS was slowly released during gastric digestion followed by rapid release during intestinal digestion. Therefore, this work prepared porous granular cold-water-soluble starch using the pulsed electric field-assisted enzymatic hydrolysis method and alcohol–alkali method, which enhanced the efficiency and cold water solubility of porous starch preparation. The PEF-P-GCWSS was a carrier with efficient zeaxanthin-loading capacity. In summary, this work expands the modification methods and applications of porous starch and provides beneficial information for further studies on the use of PEF-P-GCWSS as a lipid-soluble active substance encapsulation carrier.

In the future, more in-depth exploration is needed in the following areas:(1)The similarities and differences in the preparation of PEF-P-GCWSS from starch granules with different branched chains, different crystalline forms, and different plant sources;(2)Structural and physicochemical property changes of PEF-PS under other methods of preparing granular cold-water-soluble starch;(3)Adsorptive encapsulation of different structural guests by PEF-P-GCWSS.

## Figures and Tables

**Figure 2 foods-12-03189-f002:**
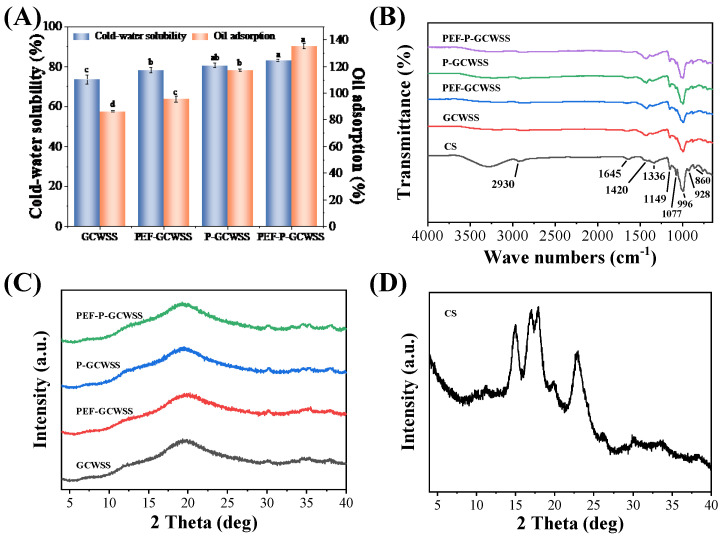
Physicochemical characterization of the GCWSS samples: (**A**) cold-water solubility and oil adsorption rate; (**B**) FTIR; (**C**,**D**) XRD. ^a,b,c,d^ values with different letters are significantly different (*p* < 0.05).

**Figure 3 foods-12-03189-f003:**
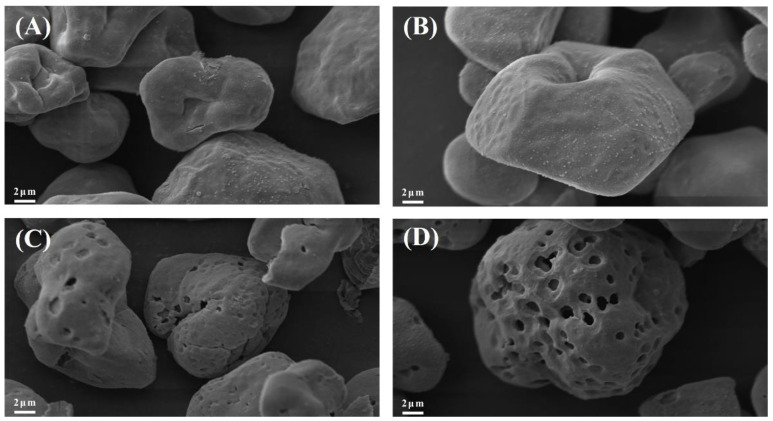
Scanning electron micrographs of different samples: (**A**) GCWSS; (**B**) PEF-GCWSS; (**C**) P-GCWSS; (**D**) PEF-P-GCWSS ×3000.

**Figure 4 foods-12-03189-f004:**
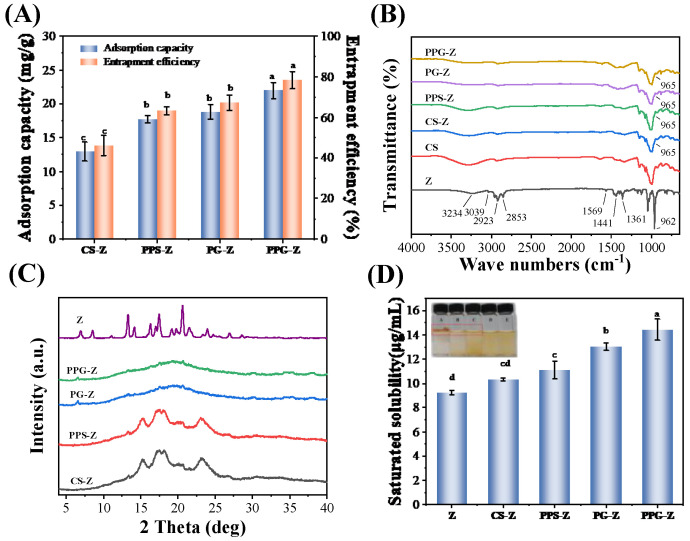
Physicochemical characterization of the GCWSS–Z samples: (**A**) zeaxanthin adsorption capacity and encapsulation efficiency; (**B**) FTIR; (**C**) XRD; (**D**) saturated solubility of the encapsulation zeaxanthin samples, the inset is a picture of the dissolved water state of different embedded samples and zeaxanthin(A: Z; B: CS-Z; C: PPS-Z; D: PG-Z; E: PPG-Z). ^a,b,c,d^ values with different letters are significantly different (*p* < 0.05).

**Figure 5 foods-12-03189-f005:**
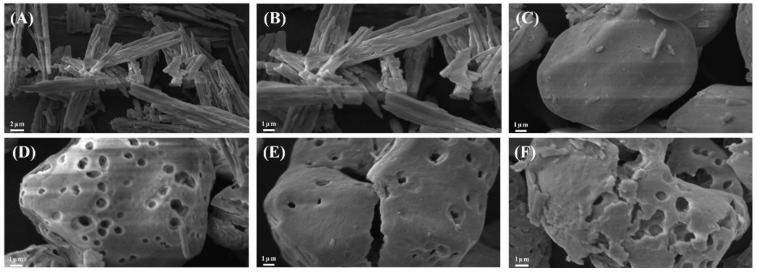
Scanning electron micrographs of different embedded samples and zeaxanthin: (**A**) Z×3000; (**B**) Z×5000; (**C**) CS–Z; (**D**) PPS–Z; (**E**) PG–Z; (**F**) PPG–Z×5000.

**Figure 6 foods-12-03189-f006:**
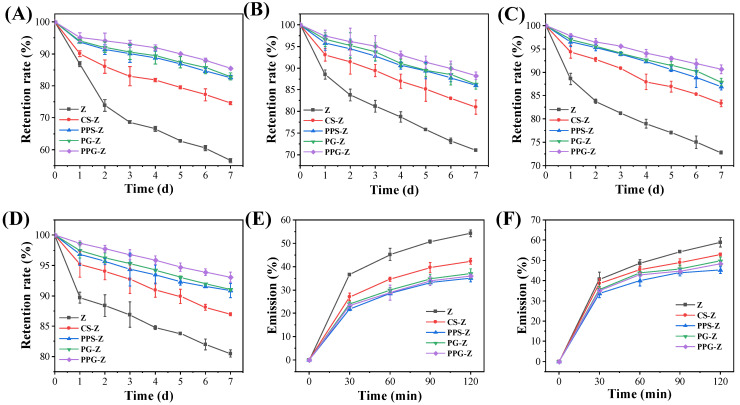
Effects of light on storage stability of the various zeaxanthin samples: (**A**) with natural light; (**B**) without light. Effects of temperature on storage stability of the various zeaxanthin samples: (**C**) 20 °C; (**D**) 4 °C. The cumulative release rates of zeaxanthin from the various zeaxanthin samples in simulated gastric/intestine conditions: (**E**) gastric; (**F**) intestine.

## Data Availability

The data used to support the findings of this study can be made available by the corresponding author upon request.

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
