# Peer review of "Pulsed Electric Field-Assisted Enzymatic and Alcoholic–Alkaline Production of Porous Granular Cold-Water-Soluble Starch: A Carrier with Efficient Zeaxanthin-Loading Capacity"

_foods, 2023, doi:10.3390/foods12173189_

Round 1

Reviewer 1 Report

After carefully reading the manuscript entitled: "Pulsed electric field assisted enzymatic and alcoholic-alkaline production of granular cold-water soluble porous starch: A carrier with efficient zeaxanthin loading capacity", it can be concluded that the authors spent a lot of time and effort in conducting experiments and writing an article. The paper is nicely written and of good quality. However, a few things could be improved. Below are remarks and suggestions.

1.     Editing of the English language, grammatic, and style is required.

2.     Some sentences are too long and difficult to understand.

3.     10% of plagiarism has been detected.

4.     Some phrases and sentences repeat throughout the text.

5.     Material and methods: Preparation of Different Starch Samples, Pulsed electric field treated corn starch (PEF-CS); After PEF treatment for a certain period of time – please define the exact time.

6.     0.5 hours – change in 30 minutes; 20 mins – change in 20 min. or 20 minutes;

7.     Ethanol concentration is missing throughout the text.

8.     Scanning electron microscopy (SEM) – insufficient data.

9.     Statistical interpretation of the results is missing and not appropriately presented. For example, differences between group means and their statistical significance were not presented.

1.     Editing of the English language, grammatic, and style is required.

2.     Some sentences are too long and difficult to understand.

3.     10% of plagiarism has been detected.

4.     Some phrases and sentences repeat throughout the text.

Author Response

Responses to reviewers' comments

Ms. Ref. No.: foods-2576932

Title: Pulsed electric field assisted enzymatic and alcoholic-alkaline production of granular cold-water soluble porous starch: A carrier with efficient zeaxanthin loading capacity

Dear Professor,

We have revised the manuscript according to the comments of the reviewers. We would like to thank the reviewers for their comments as they helped us improve this manuscript during the revision. Now, we are providing a detailed response to these comments below, we have tried to address all the comments thoroughly. Revised contents have been highlighted in red.

Thank you,

Zhong Han

Response to Reviewer 1:

General Comments

After carefully reading the manuscript entitled: "Pulsed electric field assisted enzymatic and alcoholic-alkaline production of granular cold-water soluble porous starch: A carrier with efficient zeaxanthin loading capacity", it can be concluded that the authors spent a lot of time and effort in conducting experiments and writing an article. The paper is nicely written and of good quality. However, a few things could be improved. Below are remarks and suggestions.

Response: We greatly appreciate your efforts that help us improve the manuscript.

Specific Comments

Comment 1: Editing of the English language, grammatic, and style is required.

Response: Thanks for pointing this out. Following your comment, the native speaker helped us with the language.

Comment 2: Some sentences are too long and difficult to understand.

Response: Thanks for the meaningful comment. We have modified some of the language expressions.

Comment 3: 10% of plagiarism has been detected.

Response: Respected reviewer, we have rechecked and corrected the manuscript.

Comment 4: Some phrases and sentences repeat throughout the text.

Response: Thanks for pointing this out, we have rechecked and corrected the phrases and sentences.

Comment 5: Material and methods: Preparation of Different Starch Samples, Pulsed electric field treated corn starch (PEF-CS); After PEF treatment for a certain period of time – please define the exact time.

Response: Respected reviewer, the effective treatment time of PEF treatment is 18 ms. Please check Line 149.

Comment 6: 0.5 hours – change in 30 minutes; 20 mins – change in 20 min. or 20 minutes.

Response: Revised accordingly. Please check sections 2.

Comment 7: Ethanol concentration is missing throughout the text.

Response: Respected reviewer, all ethanol of unspecified concentration in the manuscript is anhydrous ethanol. We made the corresponding modifications in the methods part.

Comment 8: Scanning electron microscopy (SEM) – insufficient data.

Response: Respected reviewer, SEM was obtained at 3000X for starch samples and 5000X for starch-zeaxanthin composites.

Comment 9: Statistical interpretation of the results is missing and not appropriately presented. For example, differences between group means and their statistical significance were not presented.

Response: Respected reviewer, we have added notes on significance in both the figure notes and the discussion section.

We appreciate your invaluable comments.

Reviewer 2 Report

Overall, it's a very informative research work with good interpretation of results. This work should attract the readers' focus. However, in the introduction section please add some points on the utilization and past records of used technologies i.e. SEM, FTIR and XRD. Is it SEM or FESEM? Can you conduct the work in FESEM as FESEM has much brighter electron source and smaller beam size than a typical SEM which increases the useful magnification of observation and imaging

Provie a section namely " Previous researches and bridge between recent trends and the past" . The "Result and discussion section: is the representative of results only. Please separate this section into two parts and provide important insights in distinguished discussion section. Include a section on "Future recommendations" after discussion section. There are some typos. Check the grammatical integrity as well.

It can be accepted after these corrections. 

Author Response

Responses to reviewers' comments

Ms. Ref. No.: foods-2576932

Title: Pulsed electric field assisted enzymatic and alcoholic-alkaline production of granular cold-water soluble porous starch: A carrier with efficient zeaxanthin loading capacity

Dear Professor,

We have revised the manuscript according to the comments of the reviewers. We would like to thank the reviewers for their comments as they helped us improve this manuscript during the revision. Now, we are providing a detailed response to these comments below, we have tried to address all the comments thoroughly. Revised contents have been highlighted in red.

Thank you,

Zhong Han

Response to Reviewer 2:

General Comments

Overall, it's a very informative research work with good interpretation of results. This work should attract the readers' focus. However, in the introduction section please add some points on the utilization and past records of used technologies i.e. SEM, FTIR and XRD. Is it SEM or FESEM? Can you conduct the work in FESEM as FESEM has much brighter electron source and smaller beam size than a typical SEM which increases the useful magnification of observation and imaging.

Provie a section namely " Previous researches and bridge between recent trends and the past". The "Result and discussion section: is the representative of results only. Please separate this section into two parts and provide important insights in distinguished discussion section. Include a section on "Future recommendations" after discussion section. There are some typos. Check the grammatical integrity as well.

Response: Thanks for your thoughtful comments that helped us improve our paper. We have responded to your valuable comments. Please reconsider our paper after reviewing the revised manuscript.

Specific Comments

Comment 1: However, in the introduction section please add some points on the utilization and past records of used technologies i.e. SEM, FTIR and XRD. Is it SEM or FESEM? Can you conduct the work in FESEM as FESEM has much brighter electron source and smaller beam size than a typical SEM which increases the useful magnification of observation and imaging.

Response: Thanks for pointing this out. Relevant descriptions of the technology were added in the introduction. Please check Lines 99-103. Due to the relatively large size of starch granules, most studies have used SEM to observe the morphology of starch.

Comment 2: Provie a section namely " Previous researches and bridge between recent trends and the past".

Response: Thanks for your thoughtful comments. We have added a paragraph to describe the between bridge recent trends and the past. Please check Lines 116-121.

Comment 3: The "Result and discussion section: is the representative of results only. Please separate this section into two parts and provide important insights in distinguished discussion section.

Response: Thanks for your thoughtful comments. We refer to the "Results and Discussion" format of most papers published in foods journals. We have added more discussion of the results. Please check Lines 297-299, 314-318, 364-366, 371-373.

Comment 4: Include a section on "Future recommendations" after discussion section.

Response: Respected reviewer, we have added a section of future recommendations at the end of conclusion. Please check Lines 542-548.

Comment 5: There are some typos. Check the grammatical integrity as well.

Response: Respected reviewer, we have checked the English language, grammatic, and style of the manuscript.

Reviewer 3 Report

Manuscript: Pulsed electric field assisted enzymatic and alcoholic-alkaline production of granular cold-water soluble porous starch: A carrier with efficient zeaxanthin loading capacity

Manuscript Number: foods-2576932

This study investigated the effect of pulsed electric field (PEF) pretreatment and alcoholic-alkaline treatment on preparation of porous granular cold-water soluble corn starch. In my opinion, this manuscript is accepted after some minor issues are addressed.

Keywords→ arrange in alphabetical order.

Materials and Methods: Add the references of Preparation of Starch and experiments.

Figures 2, 4 and 6: Don’t you have better-quality figures?

Conclusion: add insight and future application of your work at end of conclusion

Author Response

Responses to reviewers' comments

Ms. Ref. No.: foods-2576932

Title: Pulsed electric field assisted enzymatic and alcoholic-alkaline production of granular cold-water soluble porous starch: A carrier with efficient zeaxanthin loading capacity

Dear Professor,

We have revised the manuscript according to the comments of the reviewers. We would like to thank the reviewers for their comments as they helped us improve this manuscript during the revision. Now, we are providing a detailed response to these comments below, we have tried to address all the comments thoroughly. Revised contents have been highlighted in red.

Thank you,

Zhong Han

Response to Reviewer 3:

General Comments

This study investigated the effect of pulsed electric field (PEF) pretreatment and alcoholic-alkaline treatment on preparation of porous granular cold-water soluble corn starch. In my opinion, this manuscript is accepted after some minor issues are addressed.

Specific Comments

Comment 1: Keywords→ arrange in alphabetical order.

Response: Revised accordingly. Please check Line 30.

Comment 2: Materials and Methods: Add the references of Preparation of Starch and experiments.

Response: Respected reviewer, we have added the references of all methods. Please check Line 148, 167, 178, 190, 216, 223, 228.

Comment 3: Figures 2, 4 and 6: Don’t you have better-quality figures?

Response: Respected reviewer, we have uploaded the better-quality figures.

Comment 4: Conclusion: add insight and future application of your work at end of conclusion.

Response: Respected reviewer, we have added the section on future recommendations at the end of conclusion. Please check Lines 534-541.

Once again, we appreciate your invaluable comments.

Round 2

Reviewer 2 Report

Accepted in current form

English sounds ok except some minor issues